# The architecture of partisan debates: The online controversy on the no-deal Brexit

**Carlo Romano Marcello Alessandro Santagiustina**[1,2]*, **Massimo Warglien**[1]

**1** Ca' Foscari University of Venice, Venice, Italy, **2** Venice International University, Venice, Italy

☉ These authors contributed equally to this work.
* carlo.santagiustina@unive.it

**Data Availability Statement:** The dataset, containing the IDs of no-deal causal tweets, is available on GitHub at the following link: https://github.com/carlosantagiustina/DATASET_The-architecture-of-partisan-debates_The-online-

## Abstract

We propose a framework to analyse partisan debates that involves extracting, classifying and exploring the latent argumentation structure and dynamics of online societal controversies. In this paper, the focus is placed on causal arguments, and the proposed framework is applied to the Twitter debate on the consequences of a hard Brexit scenario. Regular expressions based on causative verbs, structural topic modelling, and dynamic time warping techniques were used to identify partisan faction arguments, as well as their relations, and to infer agenda-setting dynamics. The results highlight that the arguments employed by partisan factions are mostly constructed around constellations of effect-classes based on polarised verb groups. These constellations show that the no-deal debate hinges on structurally balanced building blocks. Brexiteers focus more on arguments related to greenfield trading opportunities and increased autonomy, whereas Remainers argue more about what a no-deal Brexit could destroy, focusing on hard border issues, social tensions in Ireland and Scotland and other economy- and healthcare-related problems. More notably, inferred debate leadership dynamics show that, despite their different usage of terms and arguments, the two factions' argumentation dynamics are strongly intertwined. Moreover, the identified periods in which agenda-setting roles change are linked to major events, such as extensions, elections and the Yellowhammer plan leak, and to new issues that emerged in relation to these events.

## Introduction

Online debates are a key component of contemporary democratic life, involving millions of people's expression of opinions on a vast range of topics [1, 2]. Debates are strictly associated with argumentation. Their texture is made of speech acts that express the participants' opinions and attempt to affect other participants' views by offering reasons, triggering frames and eliciting emotions [3, 4]. Debates can be finalised to deliberation, as in public assemblies, or more loosely directed to communicate and shape opinions on controversial subjects or issues. Online debates are typically of this second type. Online Social Media (OSM) platforms, such as Twitter, Facebook and Reddit, are arenas where these lively debates take place. These virtual spaces are incessantly used by wide communities of users to gather information and

controversy-on-the-no-deal-Brexit Contact information: carlo.santagiustina@unive.it.

**Funding:** C.S. and M.W. acknowledge financial support from the European Union Horizon 2020 projects ODYCCEUS (Grant Agreement No. 732942, website: www.odycceus.eu), ISEED (Grant Agreement No. 960366, website: iseedeurope.eu). C.S. also acknowledges financial support from the European Union Horizon 2020 project MUHAI (Grant Agreement No. 951846, website: muhai.univiu.org). The funders didn't play any role in the study design, data collection and analysis, decision to publish, or preparation of the manuscript.

**Competing interests:** The authors have declared that no competing interests exist.

communicate their thoughts and views concerning (realized or possible) events occurring at the national or global scale. Partisan and non-partisan participants of these online debates often publicly state their opinions about these events, their likelihood and expected effects. It has been found that social media communities constitute a non-representative sample of the population [5] and exhibit biases in terms of age, gender, and education [6]. Being predominantly made by highly educated and sociopolitically active individuals, social media ecosystems like the Twittersphere [7] reveal a wide and non-uniform spectrum of public opinions on issues of collective interest and concern [8]. While aware of the problems of generalizability and limits of Web data based research [9–11], we believe that online debates on social media platforms, being fully observable, offer a privileged window on the argumentation dynamics of a relatively large and varied sample of the politically engaged population [12], which comprises key influencers of the public sphere.

This paper introduces a methodology for analysing the argumentation patterns and structural properties of online debates and demonstrate its use by analyzing the no-deal Brexit controversy. A fundamental assumption of this work is that arguments do not come insulated in a debate, and they cannot be analysed assuming that they are independent. They are components of a multifaceted debate architecture and have to be understood in their interaction with such a structure. Public debates, such as the ones taking place on online communication arenas, are strategic discussions about socially relevant issues. At the individual level, debaters don't necessarily behave rationally but they are part of a strategic interaction. For example, they don't necessarily take into account all others' argumentation strategies, political preferences and incentives when choosing which arguments to use. However, the outcomes of their argumentation choices are interdependent from the substantial point of view, since their success (e.g., diffusion and consent) and their effects on third party opinions are generally interdependent. Therefore, interactions among debaters' argumentation choices are strategic from a substantial perspective [13]. This is true even when single arguments are seemingly chosen intuitively, in a emotionally driven way or even randomly.

Arguments can be used for a variety of purposes, like persuading an audience, orienting opinions or justifying decisions, both ex-ante or ex-post. Issues in debates always incorporate a controversial component [14], associated with partisan factions that support specific views or resolutions [15]. As debates are meant to be persuasive, they imply arguments and counterarguments. Arguments do not come alone but display different degrees of cohesiveness -they are correlated and assembled in coherent blocks. Such correlations provide structure and composite arguments to the debate. Arguments also have an internal structure and are expressed through a multitude of semantic components that offer different nuances and interpretations—these are often related to the faction expressing them. Finally, debates happen over time and are characterized by specific interaction dynamics among partisan factions, with leading and following relations, agenda-setting tentatives, attacks and defenses, shifts of dominant topics and opinions. These can be related both to the endogenous dynamics of the public debate [16, 17], such as policy communications by elected officials [18], and to external events that potentially steer them, like extreme natural events [19] or terrorist attacks [20]. Online debates are no exception: they often involve a considerably large number of participants and offer remarkable opportunities to be observed and analysed over long periods. To apprehend them, a coherent framework of observations and analysis was developed in this paper to reveal how the different architectural components of a debate are composed and linked.

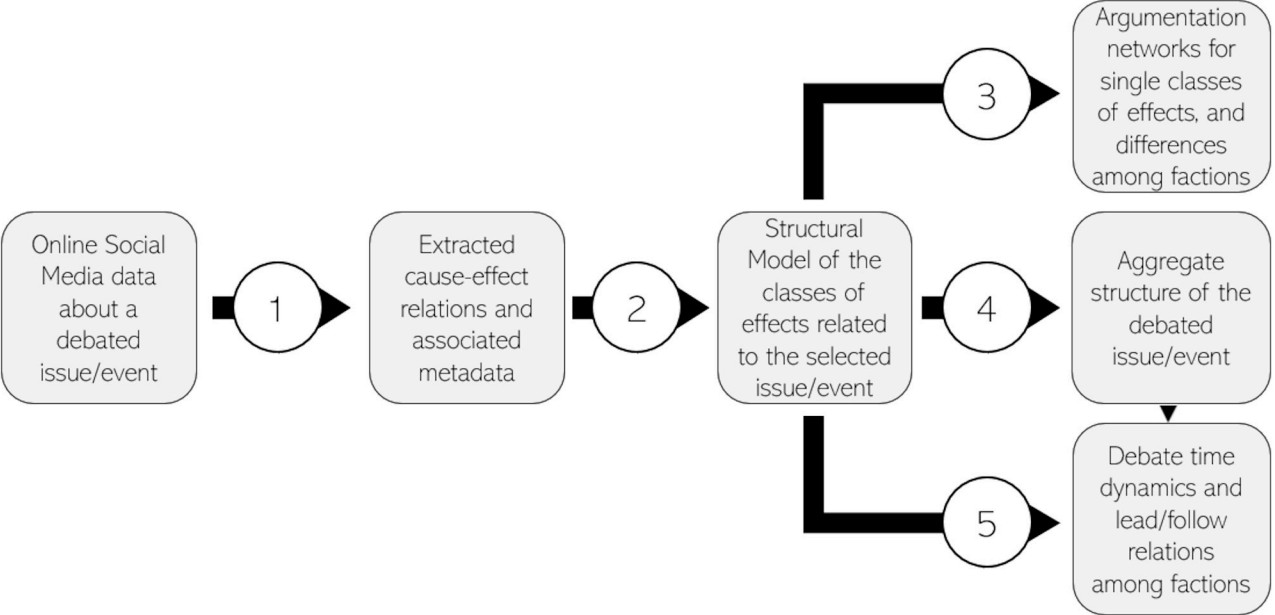

**Fig 1. Summary of the inputs and outputs of the different steps of the analysis workflow.**

## An integrated roadmap to analysing online debates

The last decade has been characterised by a sharp rise of interest concerning the use of computational methods for the automated analysis of online debates [21]. This rising interest in the subject has been accompanied by the rapid development of text mining and machine learning methods [22]. Here, a hybrid framework was developed to analyse arguments in online debates. The proposed framework builds on existing literature [23–25] by expanding and combining statistical, text mining and network analysis methods. It aims to offer a more consistent and systemic characterisation and interpretation of debates which existing tools cannot capture, and to tie together the macro-level (argumentative) and micro-level (phrasal) features of debates. In a way, our approach moves the first steps into the statistical rhetoric of (online) debates. In brief, the proposed approach can be summarized in five steps (the inputs and outputs of each step are represented in Fig 1):

1. **Argument extraction**. This paper focuses on causal arguments. Most conventional approaches [26] rely on part-of-speech recognition and relation extraction and exploit (potentially ambiguous) causal connectives to capture causal statements. Instead, the focus is here placed on verbs as causal markers which unambiguously express the semantics of causation, and for which cause-effect relations can be identified and extracted using regular expressions (RegEx). Besides being more robust for the type of OSM data employed, this method offers a rich set of possibilities in differentiating types of causal (as well as other modal) arguments [27].

2. **Aggregating causal arguments in classes (of effects)**. The second step consists in aggregating arguments via Structural Topic Modeling (STM) [28] to obtain a limited number of them. The current paper focuses on the effects which pertain to a single cause/event (i.e. the no-deal Brexit). By exploiting the metadata of the tweets and that of the extracted causal relations, differences within and between single arguments in terms of factional characterisation and types of causal relations employed are highlighted.

3. **Comparing faction rhetoric and phrases**. The internal structure of each argument was analysed through an innovative method. Each topic was transformed into an oriented, weighted graph of words and their associations using unigrams and bigrams distributions. This provides considerable additional information on how words are used and phrases are constructed inside the topic. For example, it enables us to see how analogous arguments (e.g. the economic effects of the no-deal) are differently articulated by different factions.

4. **Mapping the structure of the debate**. Arguments are correlated. The network of arguments' correlations [23] was filtered to uncover the debate's building blocks and characterise their relationships in terms of types of causal verbs, factions and inhibition/activation relationships among arguments.

5. **Identifying lead and follow faction dynamics**. The time series of the proportions of the different arguments were used to identify debate leader/follower dynamics [29] among partisan factions taking part in the debate.

The focus was placed on causal relations as a prominent example of argumentation. Although they represent a limited and very specific aspect of debates, these arguments are key for understanding (elicited) causal representations of online groups. In particular, in relation to issues of collective interest having to do with hypothetical (extreme) scenarios related to the failure in reaching an agreement among negotiating parties with conflicting interests, such as during the Brexit or the 2022 war between Ukraine and Russia. However, the proposed approach can be extended to other types of arguments, such as permission, facilitation, obstruction, possibility and influence, which are widely employed in debates [27].

## The case of a no-deal Brexit

In this paper, the proposed methodology was implemented to analyse the anatomy of the online debate on the "no-deal" hard-Brexit by extracting causal arguments, uncovering their correlation structure, and analysing the semantics of different factions' arguments. Besides, until the last extension granted by the EU Council (28 October 2019), which preceded the 2019 general elections in the UK, citizens' attitudes towards a no-deal scenario were a polarising dimension of the Brexit debate [30–32].

For analysing the "no-deal" Brexit online debate Twitter data in English published from February 2019 to May 2020 was used (see Materials and methods section for details).

By applying our methodology to Twitter posts about the "no-deal" scenario, it was possible to identify and map online arguments about the expected effects of a hard Brexit and to understand in which terms opposing partisan factions, i.e. *Brexiteers* and *Remainers*, confront each other and try to influence non-partisan online audiences through distinct argumentation and persuasion strategies.

The time frame of this work includes the first (22 March, 2019), second (10 April, 2019) and third (28 October, 2019) extensions of the Withdrawal Agreement negotiation deadline. As a result, the particular time span of this work covers the period in which the no-deal Brexit scenario became a concrete threat in the eyes of British citizens, media and politicians. Throughout the time interval between Spring 2019 and Spring 2020, the debate on social media about the possible consequences of this scenario repeatedly became more intense as the previously planned deadline became imminent and a further extension was requested by the UK government to the European Council.

## Results

### The no-deal debate on Twitter

As preliminary step, self-declared Brexit faction partisans were identified using RegEx conditions applied to the users' *Bio* field (see Materials and methods section for details). Users whose *Bio* did not match both partisan faction (*Brexiteer* and *Remainer*) conditions were considered part of a residual group, called *Others*. S4 Table in S1 File reports the number of tweets and retweets referring to the no-deal debate by faction. It was observed that both self-declared partisan factions represented around 2% of the total volume of activity. Notably, *Remainers* exhibited a higher retweet share (79,7%) with respect to *Brexiteers* (76,2%) and *Others* (75,2%).

Analysing the dynamics of the counts of "no-deal" tweets by day (see Fig 2) allowed us to single out five stages of activity associated with the phases of the Brexit process. The first stage ended in mid-April 2019, with the approval of the EU's second "flexible" extension of the UK's membership. This period was characterised by high volatility and extreme activity peaks, which generally lasted less than a week. From the end of April to the end of May, there was considerably low activity—with a rapid increase in the week before the EU Parliamentary elections. The third phase, with mid-low volumes of activity and some peaks, followed the success of the Brexit Party at the EU elections and ended with the nomination of Boris Johnson as the Tory leader. The fourth and highly active phase corresponded to the first Johnson government and ended with the third extension accorded by the EU. Finally, the subsequent period was characterised by extremely low volumes of tweeting activity about the no-deal scenario.

These phases were marked not only by different volumes of tweets but also by different levels of activity by partisan factions with respect to *Others* (see Fig 3).

The first phase was characterised by a higher initiative by *Brexiteers*, while phase 4 saw an initial burst of activity by *Remainers* followed by a more balanced debate activity of the two partisan factions.

Our approach to argument extraction is based on verbs (see Materials and methods section for details). The focus on causal verbs highlights remarkable differences in the argumentation

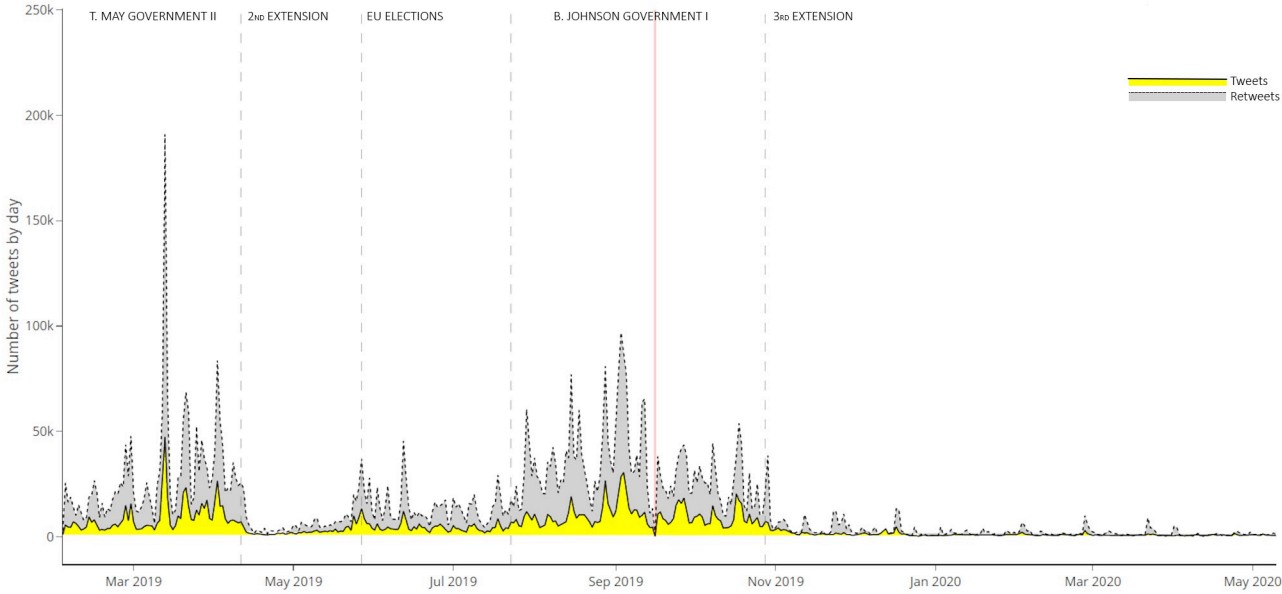

**Fig 2. Counts of the number of retrieved tweets and retweets about the "no-deal", by day.** In yellow, time series of tweets counts by day. In gray, time series of retweets counts by day. The red shaded area corresponds to 16 September 2019, server connection problems were reported on this date.

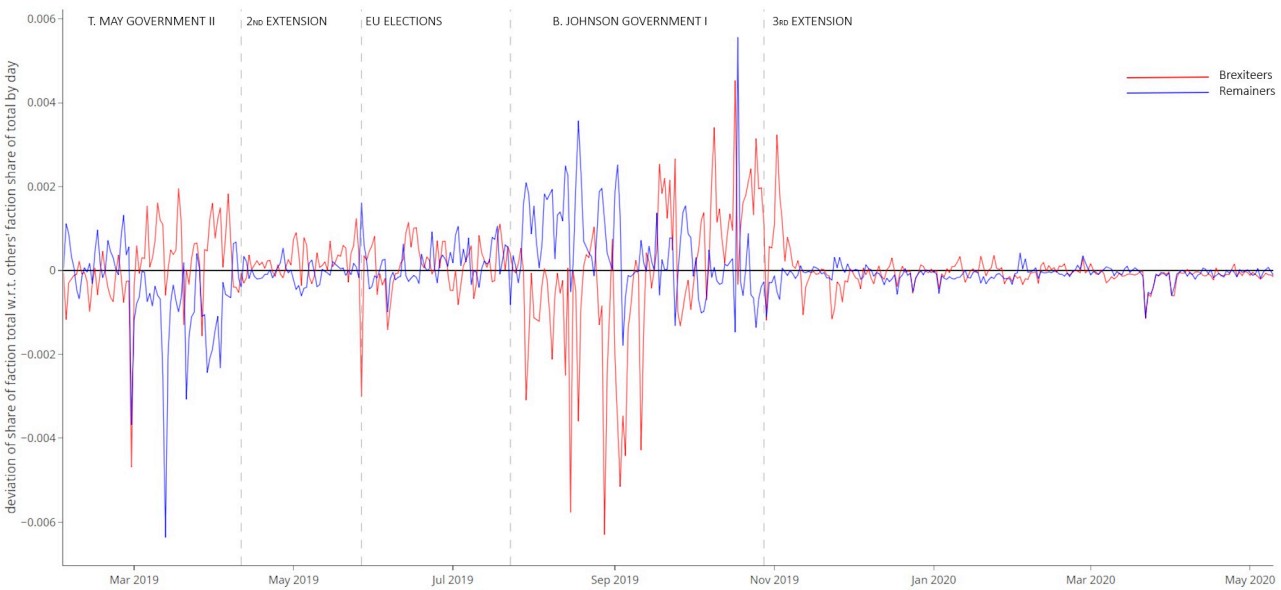

**Fig 3. Deviation of share of factions' total number of tweets about "no-deal" with respect to *Others*' faction share of total number of tweets, by day.** In red, time series of *Brexiteers*' deviation by day. In blue, time series of *Remainers*' deviation by day.

style between factions. *Remainers* use (almost 28%) more causal verbs in "no-deal"-related tweets with respect to *Brexiteers*. This difference is attenuated, but persists, for retweets (see S5 Table in S1 File). It is possible to capture semantic differences among arguments. In particular, focus was placed on three groups of causal verbs. The two groups of verbs that clearly expressed the polarity of the relation were labeled *Destruction* and *Construction*, whereas the third group, which has a neutral polarity, was labelled *Causation*. By applying our relation extraction method, ordered pairs of causes and effects were obtained for each group of verbs, with the associated metadata (see examples in Table 1).

204 648 relations were obtained, of which 36 116 contain the expression "no-deal" in the cause side of the relation. From this point onward only this set of relations was analysed. Further argumentation style differences were found to exist among factions. In particular, *Brexiteers* put stronger emphasis than *Remainers* on *Creation* relationships, while the converse is true for *Destruction* relations (see S2 Table in S1 File). This is true also when (only) the relations which are not negated are considered (see S5 Table in S1 File).

**Table 1. Examples of tweets and extracted cause-effect relations.** Tweets about the "no-deal" Brexit containing a cause-effect relation identified through RegEx, followed by extracted relation and associated metadata.

**Tweet example I**: No deal will not cause anything like the problems Remainers are predicting

| cause-side | effect-side | rel.type | negated |
|---|---|---|---|
| "No deal" | "anything [..] predicting" | causation | TRUE |

*Tweet URL*: https://twitter.com/CornockStehen/status/1162245579394539521

**Tweet example II**: Which of course rules out any free trade deal with the US since no deal will create a hard border between the two Irelands.
BBC News—Brexit bill to rule out extension to transition period https://www.bbc.com/news/election-2019-50818134

| cause-side | effect-side | rel.type | negated |
|---|---|---|---|
| "Which [. . .] no deal" | "a hard [. . .] Irelands" | creation | FALSE |

*Tweet URL*: https://twitter.com/Andrew_Hyner/status/1206934169063833600

## Aggregating no-deal arguments in classes of effects

Since extracted relations include a large and varied population of effects, there was a need to aggregate the phrases describing the effects of the no-deal in a manageable number of effect topics. This was done through a STM applied on the effect side of extracted relations. One major advantage of using structural topic modelling, is that it allows to include metadata as covariates affecting topic proportions and topic contents. This makes it possible to capture, for example, differences in how (topic contents) and how much (topic proportions) factions speak about the different expected effects of the no-deal. These differences cannot be captured with classical topic modeling techniques, such as the Latent Dirichlet Allocation (LDA) or the Correlated Topic Model (CTM). The covariates that were allowed to affect topic proportions are time (*t*), the user faction (*fct*), the relation negation (*neg*) and the relation verb group (*rel.type*). By doing so the model could be used to identify which topics characterise each partisan faction, i.e. topics that are more likely to be observed conditionally on the user belonging to one partisan faction with respect to the other. For explaining the topic content, only the user faction (*fct*) was considered. This allows to capture differences between groups, in terms of how they debate about each topic, by giving different probability weight to words (unigrams) and collocations (bigrams) that are used to express causal beliefs about the expected effects of a no-deal scenario. Fig 4 shows the 10 most probable tokens (unigrams and bigrams) for the top 20 topics, ranked in decreasing order by overall topic proportion. For example, topic 16, concerning trade agreements, (which is more likely used by *Brexiteers*) clearly shows differences in the evaluation of *Brexiteers* (stressing clean terms and the opportunities to use WTO trading agreements) and Remainers (who see the same issue as a nightmare). In topic 32 (characterising *Brexiteers*), which is about the economic consequences of the no deal, *Remainers* stress damages to jobs and the stress on the health system ("nhs" token), which disappear from the top list of words for the *Brexiteers*. Notably, for topic 28, which doesn't characterise either partisan factions and focuses on the sense (and non sense) of a no-deal scenario, the *Brexiteers* and *Others* appear to claim that a no-deal scenario will likely produce no difference (see rank of bigram token: "*no → difference*"), whereas for *Remainers* "*uncertainty*" is a more high-ranked effect. Finally, *Remainers* appear to be more concerned than *Brexiteers* and *Others* by the scenario of Scotland leaving UK as a result of a no-deal (see topic 9).

## Exploring faction rhetoric and narratives of no-deal arguments

Fig 5 shows the argumentative network of topic 2. Unigrams' probability is used to weight nodes and bigrams' probability is used to weight edges for a selected topic.

To see in which terms the two partisan factions intervene in the non-partisan debate concerning topic 2, the topic network for the faction *Others* is created, then nodes and edges were filtered separately, keeping only the 80th percentile, to prune the network from the terms (unigrams) and the collocations (bigrams) less frequently employed by non-self-declared partisan users within this topic. Subsequently, for each token in the former network, the differences in token probabilities between *Brexiteers* and *Remainers* were overlaid to the network. A red-gray-blue colour scale was used for visually representing differences. The gray colour was centered at 0 and implies no difference between partisan faction probabilities for that token (unigram/bigram). As a result, the color of edges and nodes represents the partisan faction by which a specific term or collocation is more employed in relation to the selected topic (see Material and methods section for details).

The terms "*shortages*", "*food*" and "*medicine*" are among the most relevant terms for non-partisan users (see box n.1 in Fig 5). The probabilities of these terms and collocations are significantly different for *Brexiteers* and *Remainers*. For example, while *Brexiteers* focus more on

| Topic | Topic Prop. | Brexiteers (top 10 tokens) | Others (top 10 tokens) | Remainers (top 10 tokens) |
|---|---|---|---|---|
| 24 | 4.9% | now, just, know, voting, see, happy, ever, well, fta, table | now, know, remain, sense, see, well, far, fact, last, remainers | far, now, know, right, anything, must, staff, year, saying, thinking |
| 23 | 4.5% | no-deal, borisjohnson, illegal, says, likely, no-deal->brexit, pm, scotland, says->borisjohnson, may | no-deal, likely, borisjohnson, says, pm, says->borisjohnson, likely->says, no-deal->brexit, illegal, independence | no-deal, borisjohnson, likely, pm, says, outcome, likely->says, really, perhaps, says->borisjohnson |
| 9 | 4.4% | unitedkingdom, businesses, leaving, imports, tariffs, competitive, 1, recession, outside, loss | unitedkingdom, recession, leaving, less, massive, businesses, crisis, tariffs, exports, unitedkingdom->economy | unitedkingdom, recession, leaving, massive, tariffs, living, £, unemployment, scotland->leaving, unitedkingdom->fishing |
| 16 | 4.3% | europeanunion, law, wto, deal->europeanunion, end, clean, terms, europeanunion->law, 2017, trading | europeanunion, back, law, end, come, whole, terms, working, within, deal->europeanunion | europeanunion, law, unitedkingdom->europeanunion, nightmare, europeanunion->give, backstop, neighbours, sound, end, apply |
| 20 | 4% | deal, get, leave, without, 😊, good, mps, theresamay, 😊->😊, difficult | deal, get, good, without, or, getting, negotiate, done, trying, good->deal | deal, remain, get, theresamay, best, revoke, good, done, put, travel |
| 3 | 4% | brexit, likely, happen, less, finally, wrong, talk, +, 😊, patients | brexit, want, happen, really, possible, voters, less, +, idea, less->likely | brexit, peoplesvote, voted, brexit->europeanunion, wrong, constituents, vote->no-deal, failure, happen, finally |
| 32 | 4% | economy, break, lives, world, harm, union, break->union, pound, rejoin, third | economy, damage, jobs, union, harm, lives, nhs, put, 10, risk | economy, jobs, damage, union, harm, break, nhs, huge, break->union, decades |
| 28 | 3.9% | no, sense, difference, long, no->difference, actually, no->sense, vote, default, change | no, one, sense, difference, long, power, no->sense, change, no->difference, little | no, sense, uncertainty, change, difference, harder, long, longer, no->sense, position |
| 33 | 3.8% | better, off, much, poorer, worse, better->off, threat, road, poorest, democratic | country, off, worse, much, poorer, things, difficult, worse->off, north, mean | less, poorer, off, leave, much, even, terms, lives, trade, democratic |
| 11 | 3.4% | border, northernireland, months, chaos, hard, public, ireland, land, operation, northernireland->border | border, chaos, hard, public, hard->border, ireland, months, problem, minister, northernireland | chaos, public, months, border, disorder, hard, chaos->public, months->chaos, public->disorder, hard->border |
| 26 | 3.4% | ireland, united->ireland, united, damage, government, labour, minimal, minimal->damage, labour->government, or | government, ireland, jeremycorbyn, take, northernireland, or, labour, damage, united, years | jeremycorbyn, northernireland, ireland, scotland, farming, mess, government, jeremycorbyn->government, loss->life, wales |
| 35 | 2.9% | £, going, win, give, money, millions, power, exit, little, control | money, said, going, millions, tax, harder, give, win, pay, others | money, going, benefit, £, spending, financial, tax, fortune, buddies, hide |
| 25 | 2.8% | people, want, still, believe, claiming, make, isnt, understand, thinks, die | people, still, believe, understand, national, million, die, emergency, customs, checks | people, want, still, believe, understand, lack, national, stopthecoup, make, die |
| 18 | 2.8% | years, take, sun, or, trade, minimal, government, jeremycorbyn, labour, damage | years, trade, deals, new, world, another, civil, free, trade->deals, unrest | trade, years, civil, free, another, risk, unrest, friends, civil->unrest, nation |
| 21 | 2.8% | like, look, theresamay, look->like, left, warning, fishing, brexit->fishing, fishing->warning, sacrifice->brexit, theresamay->sacrifice | like, look, theresamay, 1, look->like, thing, shit, seem, big, best | like, look, look->like, seem->like, seem, poor, fault, thing, extremely, fool |
| 34 | 2.7% | offer, voted, position, million, things, way, result, bad, 17.4, negotiation | better, way, bad, work, life, lose, option, anyone, everything, loss | better, life, worse, loss, anyone, situation, bad, plan, way, things |
| 37 | 2.7% | already, many, real, term, sides, said, deaths, mass, proper, short | many, deaths, already, term, short, shortage, hardship, mass, short->term, pain | many, deaths, misery, meds, proper, avoidable->deaths, avoidable, real, short, officials->say |
| 31 | 2.6% | 😊, economic, hit, non, year, german, economic->problems, poor, problems, no-deal->default | economic, time, say, unitedkingdom, 😊, disaster, companies, actually, hit, recession | economic, crisis, impossible, companies, utter, disruption, social, economic->crisis, norway, german |
| 15 | 2.4% | let, shortage, huge, thus, disruption, dogging, alone, water, let->alone, problems | problems, disruption, huge, 3, medicines, supply, delays, including, price, serious | disruption, medical, supplies, let, supply, cancer, drugs, medicines, increase, petrol |
| 5 | 2.4% | 50, article, article->50, party, labour, heard, no-deal, tory, default, mp | party, tory, election, labour, majority, general, must, tory->party, brexit->party, conservative | vote, party, labour, tory, tory->party, ok, changes, leader, place, general |

**Fig 4. Summary table of top 20 topics (by overall topic proportion), with top 10 tokens (by token probability) by topic and by faction.** Row color scale represents significant (at the 0.01 level) differences of the estimated topic proportion coefficients of the two partisan factions. The closer the row color is to the corresponding faction color, the more characterising is a specific topic for one of the two partisan factions. Characterising topics for *Remainers* are in blue, whereas characterising topics for *Brexiteers* are in red. For the full list of topics, refer to S6 Table in S1 File.

the term "*food*" (0.1272 *vs.* 0.0438) and the collocation "*food→ shortages*" (0.0645 *vs.* 0.0398), *Remainers* focus relatively more on the terms "*medicine*" (0.0411 *vs.* 0.0038) and "*medication*" (0.0202 *vs.* 0.0019) and the collocations "*medicine → shortages*" (0.0307 *vs.* 0.0023) and "*medication → shortages*" (0.0203 *vs.* 0.0019).

The *Remainers* also tend to focus more on the possible effects on unemployment of a no-deal scenario (see box n.2 in Fig 5); both the unigrams "*job*" (0.0566 *vs.* 0.0141) and "*losses*" (0.0283 *vs.* 0.0116) and the bigram "*job → losses*" (0.0289 *vs.* 0.0126) have higher probabilities for *Remainers* than *Brexiteers*. Finally, box n. 4 in Fig 5 shows that members of partisan

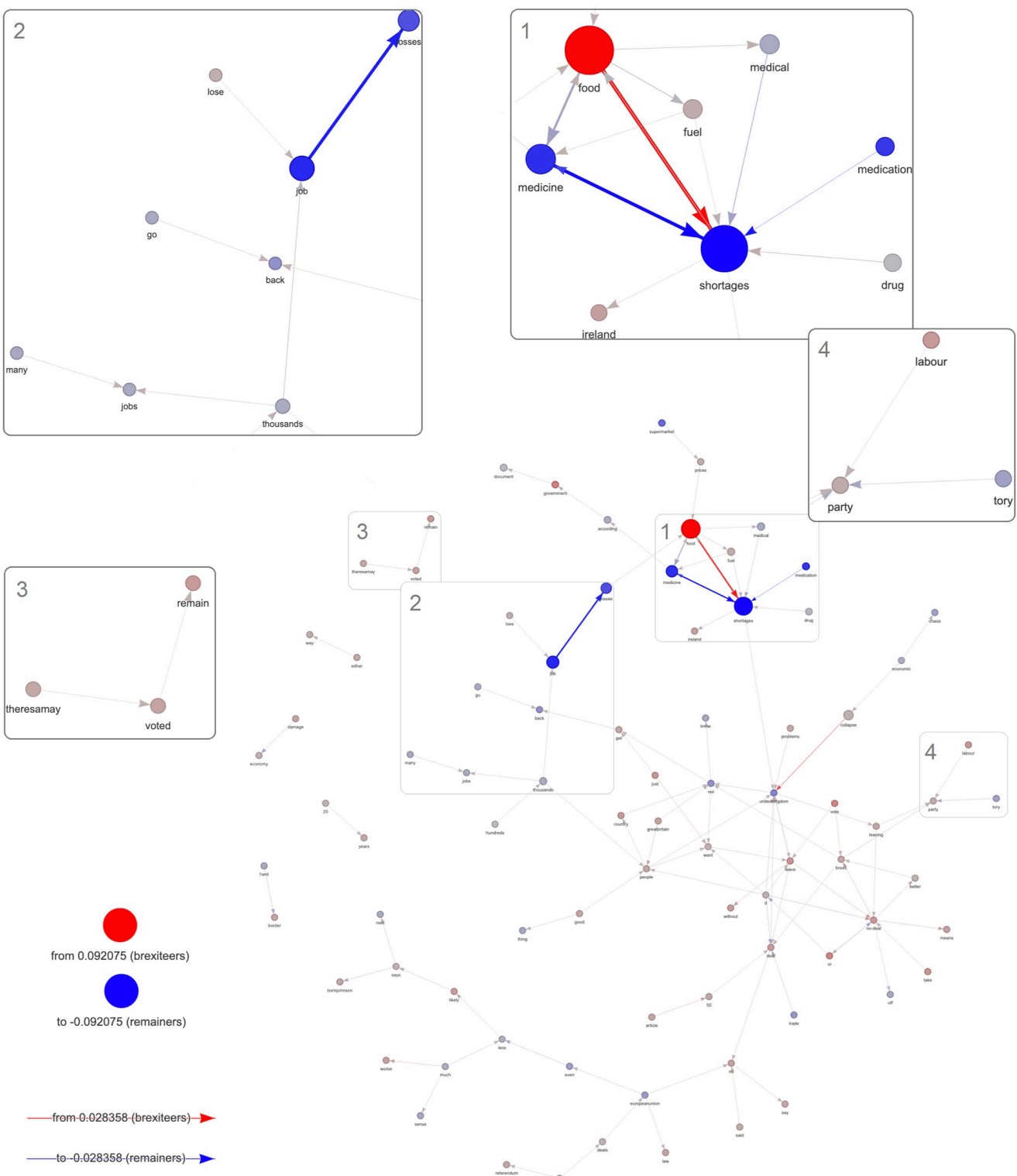

**Fig 5. Topic 2 argumentation network.** Unigrams (nodes) and bigrams (edges) have been filtered (80th percentile threshold) on their probabilities for the non-partisan faction *Others*. Node size and edge width represent probabilities of unigrams and bigrams for the non-partisan faction *Others*. Node/edge colour scales are used to represent the differences in unigram/bigram probabilities between partisan factions (*prob. Brexiteers minus prob. Remainers*). Numbered boxes display zoomed areas of interest.

factions are more inclined to talk about the party representing the other faction with respect to their own: *Remainers* use relatively more frequently the terms "*tory*" (0.00048 *vs.* 0.00026) and "*tory → party*" (0.00013 *vs.* 0.00009), whereas *Brexiteers* use more frequently the terms "*labour*" (0.00066 *vs.* 0.00011) and "*labour → party*" (0.00008 *vs.* 0.00007). This signals that while debating about the effects of no-deal scenario, partisan faction members also argue about the no-deal narratives employed by the opposing faction. The aforementioned results highlight that the proposed methodology can capture topic-specific argumentation differences between factions also at the phrasal (and phrase fragment) level.

## The structure of the no-deal debate

In a debate, classes of effects (i.e., topics) related to a common cause are not independent. Some topics likely occur together in sentences, whereas others mutually inhibit each other. For example, some effects are semantically related through the verbs expressing causal relations. Topics tend to be clustered according to verbs' polarity. Effects related to *Destruction* tend to be positively correlated among them and negatively correlated with those related to *Creation*. The same is true for *Creation* verbs. Fig 6, displaying the correlations among topics, shows the near-decomposability of causal arguments in blocks around the two diagonals of the matrix.

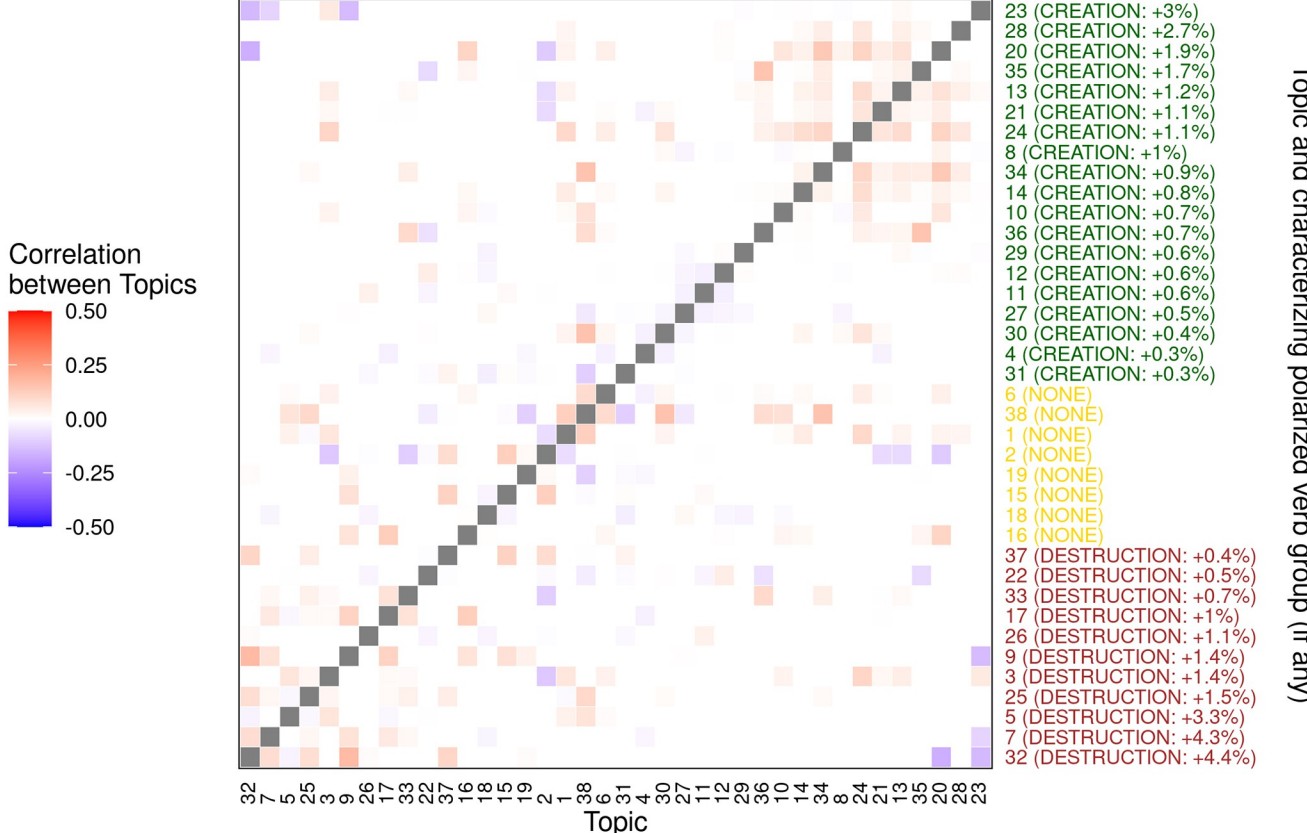

**Fig 6. Heat-map of correlations between topics.** Red colour indicates positively correlated topics and blue indicates negatively correlated topics. Non-significant correlations were identified (and set to 0) using the method proposed by Meinshausen and Buhlmann [33]. The topics were reordered on the basis of the estimated difference between the coefficients of polarised verb groups' (*Creation vs. Destruction*) effects on topic proportions $(\hat{\beta}_{k,rel=Destruction} - \hat{\beta}_{k,rel=Creation})$. If this difference is significant at the 0.01 level, the label (*Creation* or *Destruction*) of the verb group characterising the topic is contained in brackets after the topic number, followed by the value of the estimated difference (sign adjusted). If non-significant, the label *None* follows the topic number (in brackets).

The topic ordering on both axes of Fig 6 is based on topic-specific differences between the coefficients of the two polarised verb groups (*Destruction vs. Creation*), inferred with STM. Such ordering allows us to appraise to which degree arguments that are characterised by the same polarised verb group are more likely to occur together in causal tweets related to the no-deal, with respect to pairs of arguments that are characterized by different verb groups. One can capture a finer-grained structure of the debate by filtering the correlation network (see Materials and methods section for details). Filtration creates a continuum of networks resulting from the deletion of edges whose weight is below a given threshold varying over the range of observed weights. In the current setting, weights represent the value of correlations, and these are filtered according to their absolute values. At higher levels of the threshold, the network displays strong relations between arguments. The positively connected parts of the graph capture the conceptual building blocks of the debate's arguments, i.e. topic constellations. As the threshold is relaxed, a weaker (but still significant) set of relations among topics emerges and "assembles" the building blocks (see Fig 7).

Additional structural information is gained by displaying significant topic covariates in a topic correlation network and by typing the nodes by dominant partisan faction (shape), dominant polar verb group (colour) and negation of the relation (shadow), whenever the difference among types is significant (see S1 Section in S1 File). Filtration of such typed network allows us to highlight that the topic correlation network exhibits type-based assortativity, i.e. assortative-mixing of the topics for multiple (simple and composite) typing dimensions.

Fig 7 shows clear evidence of polar group verb assortativity (up to the 0.14 threshold), and partisan faction assortativity (up to the 0.10 threshold). Remark that the only pair of *Creation-Destruction* topics that is positively correlated at the filtration level 0.10 is topic 24 and topic 38. This pair of topics contains topic 24 that is one of the few topics for which the negation has a significant positive effect on the topic's expected proportion, i.e. the topic is more likely to appear conditional on the presence of a negation of the relation's verb phrase (e.g., *"[No deal] does not create [topic 24]"*).

Moreover, up to the 0.10 threshold, positively correlated triangles (3-cliques with non dashed edges) are formed by topics of non opposing types. These positively correlated triangles are also coherent in terms of polar faction types, being made either by topics that do not characterise any of the partisan factions or by topics that are not of opposing factions.

Up to the 0.10 threshold, the only triangle containing both positively and negatively correlated topics is the one with topic 23, topic 32 and topic 9, which is balanced and coherent, being formed by two positively correlated *Creation* topics (32 and 9), that are both negatively correlated with the third one that is a *Destruction* topic (23).

More generally, as the threshold was lowered to 0.05, the connected components of the graph grow respecting a basic triangle balance principle, the connected triplets of topics are always made of an odd number of positive edges. At thresholds below 0.05, most of the triangles were still balanced; however, as the threshold was further lowered, some imbalanced triangles appeared, possibly displaying the appearance of relational noise.

## The lead and follow dynamics of the no-deal debate

Debate's lead and follow relations among partisan factions are important because they reveal the capacity of a faction of setting the thematic agenda for a specific debate in a specific moment in time. Leader-follower relations among factions can be seen as processes of highly coordinated debate activity: when a faction starts discussing more or less about specific effect topics, the other faction then follows similar topic proportion variations. In the current setting, a leading faction can be seen as a synchronised group of partisan users initiating a debate shift,

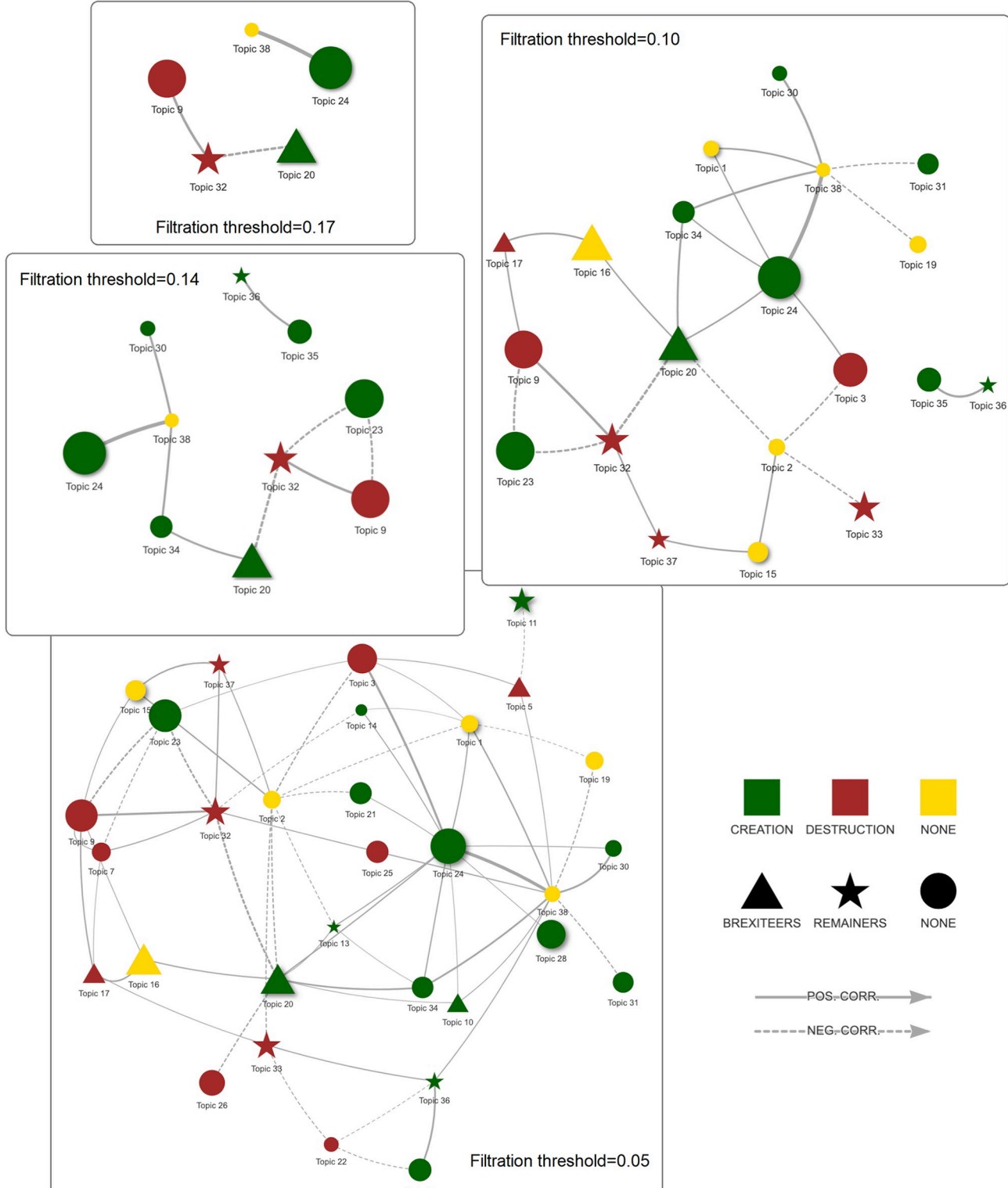

**Fig 7. Filtered topics correlation network (absolute value) and faction/type covariates.** Node color represents predominant partisan (*Brexiteer/Remainer*) faction (p.value<0.01), see legend for color details. Node shape represents predominant *Destruction/Creation* relation type (p.value<0.01), see legend for shape details. Node shadow if negated relation is predominant (p.value<0.01). Solid lines represent positive correlations whose absolute value is higher or equal to the filtration threshold. Dashed lines represent negative correlations.

whereas a following faction can be seen as a rival group of partisan users pushed to respond to the other faction by discussing in the following days the topics proposed by the latter.

To identify possibly dynamic lead-follow relations among *Brexiteers* and *Remainers*, a Dynamic Time Warping [34] method called FLICA [29] was applied. This method allows us to infer time-varying lead-follow relations between pairs of multidimensional time series, that is, between the faction-specific 38-dimensional topic proportion series, with each dimension representing the average daily proportion of a specific topic, for a specific partisan faction. We use a window of thirty (30) days, a max lag window of six (6) days, and a window time shift of one (1) day were used. Results were found to be robust to changes of the three parameters, in particular to the max lead/lag window which is the most relevant parameter representing the max range of the warping. With max warping window values from three (3) to ten (10) days, lead/follow patterns remained consistent. While, as expected, reducing the warping range mostly affected the amplitude of the observed oscillations of the index.

As we can see from the central plot in Fig 8, which shows the lead/follow index obtained by applying FLICA, lead-follow relations among the two partisan factions exhibit multiple cycles with different intensities and degrees of persistence. In particular, by relating the peaks and troughs of the index to political events in the UK, it is noticed that peaks correspond to moments in which politicians and political parties close to *Brexiteers* were in a position of

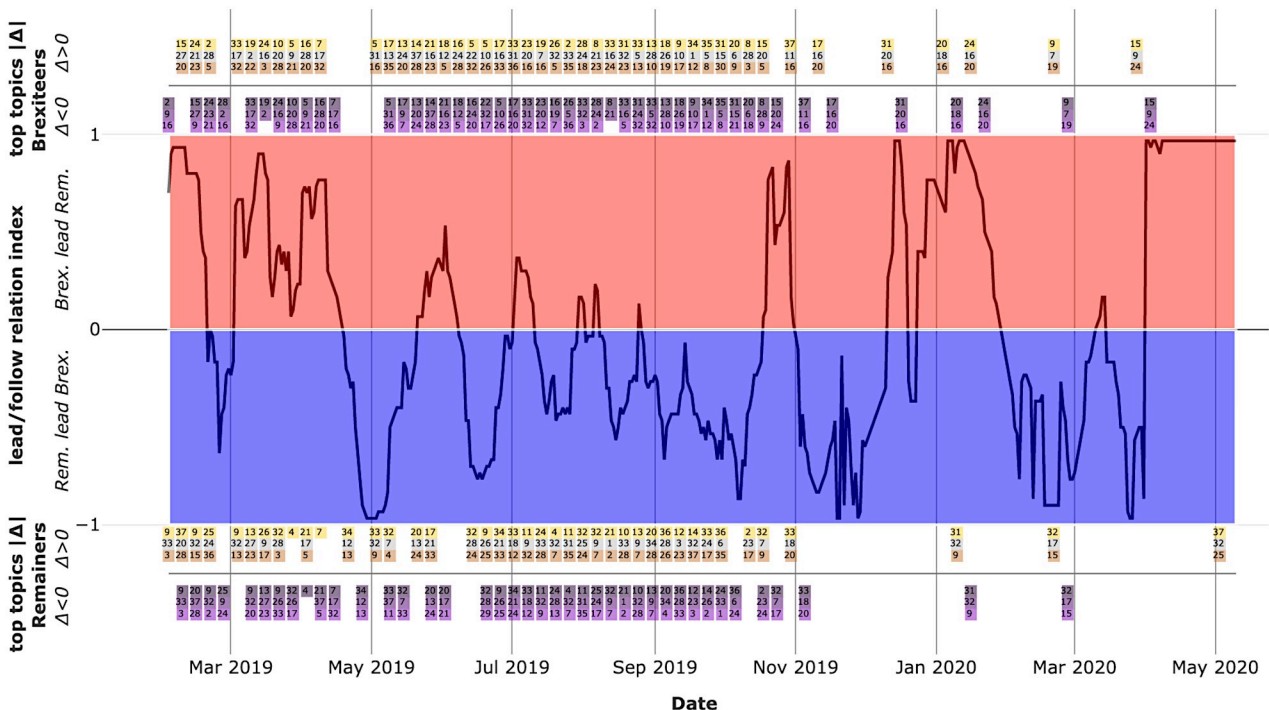

**Fig 8. Faction lead/follow relations.** *Central plot*: lead/follow relation index among partisan factions. The index was constructed using the method proposed by [29], and it is based on Dynamic Time Warping of the 38-dimensional daily topics proportions series by faction, using a time window of thirty (30) days, a max lag window of six (6) days, and a window shift of one (1) day. The closer the index value is to one (1), the more intense is the lead of *Brexiteers* on *Remainers* on that day, and the converse is true for values close to minus one (-1). *Upper and lower plots*: Top topics by absolute weekly topic-proportion variation $|w_{x,k,\,i} - w_{x-7,\,k,\,i}|$ (see Materials and methods section for details). Inside gold, silver and bronze boxes are contained the IDs of the top three topics, in terms of largest weekly positive variations, representing respectively the 1st, 2nd and 3rd ranked topics. Inside the dark violet, violet and light violet boxes are contained the IDs of the top three topics in terms of largest weekly negative variations, representing respectively the 1st, 2nd and 3rd ranked topics. Some boxes may be missing when non-null (positive/negative) weekly topic proportions changes are observed for less than three topics.

strength, whereas the converse is true for troughs. Notably, rapid shifts in leadership occurred after the following events:

- **MPs Amber Rudd ultimatum (22 February 2019)**: *Brexiteers* led the debate in the first two decades of February and then *Remainers* took the lead until the beginning of March 2019.

- **2nd extension (10 April 2019)**: *Brexiteers* led the debate from the beginning of March to mid April and then *Remainers* took the lead until the second week of May 2019.

- **EU elections (23 May 2019) and the resignation of Theresa May (7 Jun. 2019)**: *Brexiteers* led the debate from May 21st to the day on which T. May's resignation became effective.

- **Yellowhammer plan leak (18 August 2019)**: *Remainers* led the debate from mid August until mid October 2019 (excluding 25 August 2019).

- **2019 United Kingdom General Election (12 December 2019)**: *Remainers* led the debate from the beginning of November until two days before the GE and then *Brexiteers* took the lead until the end of January 2020 (excluding 21–23 December 2019).

Moreover, by jointly analysing the lead/follow index and the weekly topic variations (see upper and lower plots in Fig 8), some highly significant patterns were observed. First (i), periods where *Remainers* lead or where leadership starts shifting towards them are often characterised by the presence among the top three topics (by weekly positive-sign variation) of topic 32 and topic 9, which are both related to the economic effects of a no-deal scenario. These two topics are also dominated by *Destruction* causal verbs and form one of the strongest macroargument components identified in Fig 7. Moreover, topic 32 was also found to be characterising for *Remainers*. Second (ii), periods where *Brexiteers* lead or where leadership starts shifting towards them are similarly characterised by the presence among the top 3 topics (by weekly positive-sign variation) of the topic 20 and topic 24, which are both dominated by *Creation* causal verbs, and trivialize the alternative to a no-deal scenario by ridiculing the difficulties and delays in the negotiations of a trade deal with the EU. Topic 20 characterised *Brexiteers* and contains several emojis that jokingly refer to the extremely remote possibility of reaching an agreement and hence avoiding a no-deal scenario.

Both points suggest that the structural properties identified in the effect topic-correlation network analysis also play a role in the dynamics of lead/follow relations among factions—further supporting the importance to consider the interdependence of arguments and of levels of analysis.

## Discussion

The present work shows that the structure and dynamics of the debate bind arguments in a multi-level "architecture". This architecture integrates the analysis of interdependent arguments, their factional specificities, and leading/following roles of partisan factions. Moreover, the current reconstruction of causal arguments in the no-deal debate allows to unveil how the debate dynamics relate to external events and how they are shaped by factional interests. For example, *Remainers* resort more extensively to causal arguments than *Brexiteers* do, they emphasise the potential destructive causal effects of the no-deal Brexit, and stress its (negative) economic implications.

In particular, it was shown that the network of no-deal causal arguments displays polarised assortativity around structurally balanced building blocks that aggregate topics by (verb) type and factional orientation. Correlated constellations of arguments also play an important role

in the debate dynamics, as they often mark, in conjunction with external events, shifts in the factional leadership of the debate.

Thus, better insights can be obtained by addressing the complex architecture of debates through approaches that combine different tools in a coordinated way. In this paper a multi-step approach is explored. This approach traces a methodological roadmap through the different architectural components of the debate—pointing in the direction of the development of a statistical rhetoric of debates. To this end, we had to adapt pre-exisiting tools in innovative ways, e.g. by combining structural topic modeling with network modelling. This allows to ask new questions about the structural properties of debates that generalise beyond the specificity of the current case study. For example, the filtration of the correlation network of effect topics suggest some notable structural regularities—e.g., argument triangles are all structurally balanced until very low levels of correlation (possibly corresponding to relational noise) are considered.

Besides this general contribution, this paper further contributes to more specific streams of literature. Our work clearly has roots in the pioneering work of Axelrod on cognitive maps [35] and in the developments of network text analysis [36, 37]. As in [35], our focus is on extracting causal arguments, and as in network text analysis we show how concepts are represented by networks of words in different social aggregates. However, the proposed set of tools used to extract the architecture of argument systems in debates focuses on arguments' mutual interrelation and their factional and temporal dynamics. The current work also resonates with the recent attempt to unveil variation in patterns of topics distribution in single speeches and its impact in debates [21]. Rather than considering speeches as weighted combinations of independent topics, this approach looks at the interdependence of topics in texts, uncovering deeper levels of structure in debates.

In relation to Structural Topic Models, this work highlights how classification algorithms that jointly exploit text and metadata can be fruitfully used not only with short online social media posts, such as tweets [38], but also with subsets of short posts which may represent phrases or causal relations extracted with RegEx or NLP algorithms from the former. Moreover, besides using classical document-level covariates, such as the author and the publishing date, this work shows that one can transform otherwise non-exploited textual data from a post or from its metadata in valuable categorical covariates, such as the faction and causal relation verb type.

From the point of view of network science, this paper demonstrates that from the micro to the macro scale, graphs rather than sets, appear to be the most appropriate way to analyse and model debates: At the micro level, the inclusion of bigrams in the modelling setting allowed us to reconstruct faction-specific argumentation networks, which highlight in which terms single factions intervene in a topical debate; at the macro level, the combination of the topics' correlation matrix with predominant covariate-level labels allowed us to model the architecture of the debate, and identify its salient dimensions and topological properties. Differently from [39], which built a bipartite network of documents and words and extracted topics as communities, in this work combined structural topic modelling techniques and network representations, to enable a systemic analysis of arguments in online debates. From a broader perspective, this work contributes to the emerging field of narrative economics [40] and constructivist approaches to socio-economic issues [41] by offering an extensive framework for studying online causal debates and opinion dynamics.

The present analysis of the no-deal debate considers only causal arguments that have a single cause (the "no-deal" causal factor) and focuses on different effects associated to it. This constraint was introduced to keep the analysis simple enough, however it is not a an intrinsic limitation of our approach. Indeed, one may consider multiple causes o a single effect or

multiple causes of multiple effects as well. A second limitation is that the phrasal networks were reconstructed only within each topic (i.e., effect class). However, as shown, the arguments are correlated and form detectable constellations of topics. It is possible to reconstruct phrasal networks for such constellations as well. This would potentially bring to light more connections among words and the phrasal constituents of more complex arguments and narratives. Third, more systematic information could be extracted by looking at structural indicators of networks, such as centrality, betweenness, modularity and community structure. Such issues have been explored in a separate paper on COVID-19-related literature [42]. Finally, in the current analysis retweets, which provide natural indicators of the social resonance of arguments, were not considered. As this deviates from the main focus of this paper, this analysis was deferred to a future work.

## Materials and methods

The data acquisition, preparation and modelling procedures used in this study are as follows.

### Data collection and pre-processing

Tweets about the no-deal Brexit were collected through Twitter's Stream API, from the beginning of February 2019 to beginning of May 2020. Twitter's Stream API endpoint has been queried through the DMI-TCAT [43] open-source software, using as stream filters "no deal", "no-deal" and "nodeal". In total, more than 9 million tweets were downloaded and archived in an SQL database. The downloaded data were then pre-processed to remove tweets that are unrelated to the no-deal Brexit. In particular, tweets related to the US-China trade war (and its no-deal scenario) were identified and hence removed using a RegEx containing references to both countries or their leaders.

The cleaned data-set contains 9004927 tweets about the no-deal Brexit and their metadata, many of which are retweets, which we exclude from the next step of our analysis.

### Arguments and argument-specific covariates extraction

To find and extract from the no-deal tweet strings of text that identify cause-effect relations, a RegEx algorithm that exploits verbs was employed. In particular, a list of verbs and verb phrases related to *Causation*, *Creation* and *Destruction* (see S11 Table in S1 File) was used to build a set of RegEx functions, which were used to identify and separate the two sides of each cause-effect relation contained in the tweets. The steps of the the cause-effect extraction process can be summarised as follows (for RegEx algorithm details and commented code we refer to S12 Table in S1 File):

1 Tweets were segmented in sentences using punctuation characters (the Quanteda library [44] was used for this purpose).

2 For each sentence:

 2.1 Verb phrases related to cause-effect relations were identified through RegEx functions.

 2.2 If there was one or more RegEx matching, the following elements of the verb phrase were identified and recorded:

 2.2.1 position (*start char.* and *end char.*);

 2.2.2 type (*causation*, *creation*, or *destruction*);

 2.2.3 negation (e.g., *"Y will **not** cause X"* implies *neg = TRUE*) if any;

2.2.4 verbal form, which can be:

- active (e.g., *"X will cause Y"*);

- passive (e.g., *"Y will be caused by X"*);

- end-of-sentence (e.g., *"Y that X will cause"*);

2.3 For each sentence with at least one RegEx matching, the previously extracted information was used together with a set of verbal form specific functions to split (and reorder) the different components of each sentence in one ore more cause-effect relation triplets, each of which included the following elements (examples that follow are based on this sentence: *"A no-deal Brexit would certainly destroy UK's economy and labour market"*):

- a subject representing the *cause-side* of the relation (e.g., *"A no-deal Brexit [. . .]"*);

- a predicate representing the *relation type* to which the verb phrase corresponds (e.g., *"[. . .] would certainly destroy [. . .]"* → rel.type = *Destruction*);

- an object representing the *effect-side* of the relation (e.g., *"[. . .] UK's economy and labour market."*).

We hence obtain a set of cause-effect relation triplets. Each relation triplet $i$ is also associated to the following set of covariates:

- $t_i$—the date-time of the relation $i$ (which is the publishing date-time UTC of the tweet from where the relation $i$ was extracted)

- $rel.type_i$—the verb phrase type of the relation $i$ (3-levels categorical: *Creation*, *Causation*, *Destruction*)

- $neg_i$—a verb negation dummy for the relation $i$ (binary):

  - $neg_i$ = *TRUE* if the verb phrase of the relation $i$ contains a negation;

  - otherwise $neg_i$ = *FALSE*;

- $fct_i$—the faction of the author of the relation $i$ (3-levels categorical: *Brexiteer*, *Remainer*, *Other*):

  - $fct_i$ = *Brexiteer* if the biography of the user that has posted the tweet from which the relation $i$ was extracted matches at least one -case insensitive- RegEx condition contained in the *Brexiteer* dictionary ($dict_{Brexiteer}$ = {*brexiteer, vote brexit, voted brexit, voted for brexit, ukip, brexit party, vote leave, leave the EU, respect my vote, johnson, farage, anti-eu, antieu*}) and none of the conditions contained in the *Remainer* dictionary ($dict_{Remainer}$ = {*remainer, vote remain, voted remain, voted for remain, remain party, new vote, stay in the eu, pro-eu, proeu*});

  - $fct_i$ = *Remainer* if the biography of the user that has posted the tweet from which the relation $i$ was extracted matches at least one -case insensitive- RegEx condition contained in $dict_{Remainer}$ and none of those contained in $dict_{Brexiteer}$;

  - otherwise $fct_i$ = *Other*.

If on one side, the values of the covariates *rel* and *neg* were inferred through RegEx functions applied to the tweet's text (i.e., the Twitter post's content), on the other, the faction covariate *fct* was inferred from the biographical information of the users. Hence, in this work, *Brexiteers* and *Remainers* are those users that self-identify with one of those factions

and openly declare it in their *Bio* on Twitter. Even though this condition is rather stringent, it allows to minimise the risk of including "false positives" in our two partisan factions. Besides, being more demanding than other methods based on retweet and following networks, our partisan self-identification method allows for faction changes: a tweet (and the cause-effect relations therein contained) was considered to have been posted by a partisan user if (and only if), at the moment the tweet was posted, the bio of the user posting it matched one of the two partisan faction dictionary conditions. As a result, this framework allows Twitter users that take part to the no-deal debate to dynamically enter and exit a partisan faction, as the Brexit debate and the self-declared faction of users taking part in it change across time.

## Aggregating arguments and estimating covariate effects

Through the previous steps, a set of 204648 relations was obtained, each containing a cause side and an effect side, hereinafter simply called *cause* and *effect*. Since this work focuses on the declared effects of a no-deal scenario, all extracted relations whose *cause* did not match specific RegEx conditions used to verify the presence of *no-deal* in the subject of the relation were filter out (see S3 Table in S1 File for details). A set of 36116 relations that matched the aforementioned RegEx was thus obtained.

To aggregate the extracted no-deal effects in classes of effects (i.e., topics) and to see in which terms the propensity to speak about these classes may depend on covariates, a STM was estimated using only the previously extracted 36116 effects related to the no-deal. STM was selected as topic modelling technique for its unique combination of features required to fulfill the objectives of this work. (i) First, being an extension of the Dirichlet-Multinomial Regression topic model [24], STM allows for the inclusion of covariate information in the estimation process. This affects the estimation through informative priors and, more importantly, allows to infer the effects of extracted covariates on topic proportions. (ii) Second, being constructed upon the Correlated Topic Model [23] it allows us to infer the interdependence structure among topics that co-occur in (the effect-side of) relations that have no-deal inside the subject of the triplet. (iii) Finally, being a generalization of the Sparse Additive Generative [25] topic model, it allows covariates to affect the contents of a topic, through sparse deviations with respect to a baseline distribution. This feature was applied in this work to the *fct* covariate, to model and analyse in which terms, for a given topic, faction-specific argumentative styles can be distinguished from one another.

Each effect $d \in \{1, \ldots, D\}$ (where $D = 36116$) is represented as a set of tokens from a vocabulary of unigrams and bigrams, indexed by $v \in \{1, \ldots, N\}$. Effects were hence transformed in a matrix called **ExT** of size $D$−by−$K$ containing the counts of the number of tokens by effect. As a modelling strategy, covariates contained in the $(D)$−by−$(4)$ matrix $\mathbf{X} = \{t, rel, neg, fct\}$ were allowed to affect topic proportions, whereas only the covariate vector $Y = \{fct\}$ was allowed to affect the contents of topics. The choice of having the faction covariate *fct* affect contents is related to the objective of understanding if and in which terms partisan factions taking part in the Brexit debate use different words (unigrams) and associations (bigrams) to speak about an inferred class of no-deal effects (i.e., topic). As tokens, only unigrams and bigrams which appear at least 10 times in the final collection of no-deal effects were included. This resulted in a vocabulary $V$ made of $N = 3505$ tokens, of which $N_u = 2462$ are unigrams and $N_b = 1043$ are bigrams.

The matrices **ExT** and **X** were used, together with the vector $Y$, as inputs to estimate our model using the *Stm* package for R [45].

A STM with $K$ topics is defined as follows:

**Topic proportion**

$$\mu_{d,k} = X_d \gamma_k$$

$$\gamma_k \sim \mathcal{N}\left(0, \sigma_k^2\right)$$

$$\sigma_k^2 \sim \text{Gamma}(s^\gamma, r^\gamma)$$

(1)

**Language model**

$$\theta_d \sim LogisticNormal(\mu_d, \Sigma)$$

$$z_{d,n} \sim \text{Mult}(\theta_d)$$

$$v_{d,n} \sim \text{Mult}\left(\beta_d^{k=z_{d,n}}\right)$$

(2)

**Topic content**

$$\beta_{d,v}^k \propto \exp\left(m_v + \kappa_v^{\cdot,k} + \kappa_v^{y,\cdot} + \kappa_v^{y,k}\right)$$

$$\kappa_v^{y,k} \sim \text{Laplace}\left(0, \tau_v^{y,k}\right)$$

$$\tau_v^{y,k} \sim \text{Gamma}(s^\kappa, r^\kappa)$$

(3)

Where topics are indexed by $k$, $X_d$ is a 1–by–4 vector, $\gamma_k$ is a 4–by–$K$ matrix of coefficients, and $\Sigma$ is a $K$–by–$K$ topic proportion covariance matrix. The distribution over tokens is the combination of three effects: a topic effect ($\kappa_v^{\cdot,k}$), a *fct* covariate effect ($\kappa_v^{y,\cdot}$) and a topic-covariate interaction effect ($\kappa_v^{y,k}$). These three effects are modelled as sparse deviations from a baseline token frequency ($m_v$). To choose the number of topics $K$, the model was estimated for different values of $K$ ranging from 3 to 70. For each value of $K$, the following procedure was repeated 50 times: (i) split the corpus in a random training set and a test set (the training set contains a random sample containing 25% of the total number of no-deal *effects*) using a different random seed at each repetition; (ii) estimate the STM model (see S2 Section in S1 File for details about STM parameter values) and (iii) compute the lower bound and the mean likelihood of the STM to evaluate its performance. Then for each $K \in \{3, ..., 70\}$, the average values of the lower bound and the mean likelihood were computed. The aforementioned model performance indicators suggested that $K = 38$ was a good candidate number of topics (see S2 Section in S1 File). Finally, the STM was re-estimated for $K = 38$ with the whole set of *effects* using spectral initialization, which allows the estimated model to be deterministic conditionally on parameters and covariates values.

## Constructing faction-specific narrative networks

To construct faction specific narrative networks for a topic $k$, we use the posteriors of $m_v$, $\kappa_v^{\cdot,k}$, $\kappa_v^{y,\cdot}$ and $\kappa_v^{y,k}$, which are respectively called $\hat{m}_v$, $\hat{\kappa}_v^{\cdot,k}$, $\hat{\kappa}_v^{y,\cdot}$ and $\hat{\kappa}_v^{y,k}$. In particular, tokens (i.e., unigrams and bigrams) in the vocabulary $V$ were separated into two disjoint sets $V_u$ and $V_b$, where $V_u$ contains only the unigrams from the vocabulary $V$ and $V_b$ only the bigrams. For all token $v \in V_u$, the less relevant unigrams for the $k$-th topic and for faction *Others* were filter out. This was done by keeping only tokens above the 80*th* percentile rank, in terms of the following posteriors sum: $\hat{m}_v + \hat{\kappa}_v^{\cdot,k} + \hat{\kappa}_v^{y=Others,\cdot} + \hat{\kappa}_v^{y=Others,k}$. The same procedure was then applied for bigrams (all $v \in V_b$). Finally, bigrams that did not connect unigrams pairs in the 20% top percentile were filtered out. From the resulting unigram and bigram sets and their weights,

which are given by the exponential of the aforementioned posteriors sum, we constructed the narrative network of the topic $k$ for non-partisan users ($fct = Others$). This was done by using unigrams as nodes and bigrams as edges, and by representing unigrams' weights through the node size and bigrams' weights through the edge width. The resulting network can be seen as a graphical representation of the phrasal microstructure of the debate about a topic, for non-partisan users. Since the objective was to analyse how partisan factions intervene in this debate, the topic-specific token probability differences between the two partisan factions was overlaid to the network using a continuous color scale ranging from blue (for negative values) to light-gray (for zero) to red (for positive values). For any token $v$, the posterior probability difference between faction $i$ and faction $j$ is given as follows:

$$\hat{\delta}_{v,i,j} = exp(\hat{m}_v + \hat{\kappa}_v^{y=i,} + \hat{\kappa}_v^{y=i,k}) - exp(\hat{m}_v - \hat{\kappa}_v^{y=j,} + \hat{\kappa}_v^{y=j,k}) \tag{4}$$

When $i = Brex.$ and $j = Rem.$, we obtain the difference between the *Brexiteers* and *Remainers* partisan factions for topic $k$. Colors representing partisan factions differences are then overlaid to the narrative network of non-partisan users (i.e., *Others*), as shown in Fig 5 for topic 2.

## Filtering the network structure of a debate and identifying constellations of effect-classes

To explore the relationship between covariate values and topic proportions, the *estimateEffect* function of the STM library was used. This function allows to infer the effects of one or more covariates included in the STM on expected topic proportions. For each *effect d*, the proportion of any topic $k$ was modelled as a function of the faction of the author of the post containing the effect $d$ ($fct_d$), its relation type ($rel.type_d$), and a dummy representing verb negations ($neg_d$):

$$propensity_{k,d} = f(fct_d, neg_d, rel.type_d) \tag{5}$$

This method allowed us to asses which covariate coefficients are statistically significant (see S7 Table in S1 File for regression results).

To analyse if there are significant differences in topic prevalence among the two partisan factions (*Brexiteers vs. Remainers*) and among the two polarised verb types (*Creation vs. Destruction*), the coefficients' differences and variances were computed to test if the former were statistically different.

To represent the aggregate structure of the debate about no-deal Brexit effects, the $\hat{\Sigma}$ matrix, containing inferred correlations between pairs of topics, was transformed in a topics propensity correlation network that can be visualised as an undirected graph, where nodes represent topics and edges represent correlations between them. This topics correlation network was then labelled on the basis of covariate-levels that have predominant effects (i.e., significantly larger coefficients) with respect to their opposing type (*Brex. vs. Rem.*, *Creation vs. Destruction*). More specifically, for each topic, the corresponding node was labelled on the basis of the covariate level that implied a significantly higher propensity for that topic (if any), otherwise the predominance relation property was labelled with the "none" label. The topic's predominant relation type ($\hat{\beta}_{rel.type=Destruction}$ vs. $\hat{\beta}_{rel.type=Creation}$) is represented through the node color, whereas, the topic's predominant partisan faction ($\hat{\beta}_{rel.type=Brex.}$ vs. $\hat{\beta}_{rel.type=Rem.}$) is represented through the node shape. Finally, significant positive effects on a topic's propensity related to the presence of a negated causal verb (i.e., $\hat{\beta}_{neg=TRUE} > 0$) were represented by applying a shadow around the topic node. To highlight the structural relations among arguments employed in the no-deal effects debate and identify topic constellations that

attract or repulse each other, the correlation graph was filtered using different threshold levels [46]. Filters where applied to the absolute value of the correlations, which are represented through the edges' width. These threshold values were progressively lowered (in absolute terms), and at each step isolated nodes were removed to show only the backbone of the debate for that specific filtration threshold. This allows to analyse the building blocks of a debate and how these blocks grow as we lower the filtration threshold. Moreover, by analysing the topology of this network (e.g., balanced and unbalanced triangles or cliques) [47] one can see if these building blocks are coherent either in terms of the sign of the correlations that characterise them, or in terms of the property labels associated to predominant covariate levels.

## Identifying time-varying faction lead/follow relations

Many methods to analyse lead-follow relations among time series exist, like cross-correlations among faction-specific daily topic proportions series. Despite their usefulness, these methods have several limits, in particular, the resulting lead-follow relations cannot change across time. As a result, a method based on Dynamic Time Warping [34] (DTM) was applied in this work. This method allows to infer lead-follow relations among factions that may change across time. In brief, this approach allows us to model factions as groups of agents moving, from day to day, in an argumentation space, sometimes following each other and other times diverging. As a first step to identify the time-varying faction lead-follow relations, the $D-$by$-K$ matrix containing the distribution of topics by effect, called $\mathbf{ExT}$, was extracted from the estimated STM. $\mathbf{ExT}_{d,k}$ represents the estimated propensity of topic $k$ in the no-deal *effect* $d$, and $\sum_{j=1}^{K} \mathbf{ExT}_{d,j} = 1$. Using $\mathbf{ExT}$ together with the $t$ and $fct$ covariates contained in $\mathbf{X}$, for each faction $i \in \{Brex., \ldots, Rem.\}$, for each topic $k \in \{1, .., K\}$, and for each day $x \in \{01 - 02 - 2019, 02 - 02 - 2019, \ldots, 01 - 05 - 2020\}$, the average daily propensity of topic $k$ for faction $i$ on day $x$ was computed and called $w_{x,k,\,i}$. Where for a specific day $x$ and faction $i$, we have that $w_{x,k,i} > 0 \; \exists k$ and $\sum_j w_{x,j,\,i} = 1$ if there is at least one effect $d$ that has $x$ as date ($t_d$ covariate equal to $x$) and $i$ as faction ($fct_d$ covariate equal to $i$), and $w_{x,k,\,i} = 0 \forall k$ otherwise ($w_{x,k,i} = 0 \forall k \Rightarrow \sum_j w_{x,j,\,i} = 0$). Each matrix $w_{.,.,i}$ is of size $T-$by$-K$ and contains as column vectors $K$ time-series with the average estimated topic propensities of extracted effects posted by users belonging to faction $i$. Dynamic lead-follow relations among factions were identified with a DTM algorithm called mFLICA [29]. In this framework, the notion of leading entity (i.e., leading faction) corresponds to the initiation of topical proportion patterns that other factions hence follow. Given a set of time series representing average daily topic proportions in documents by faction, one can use this method to identify periods of coordinated activity between factions and infer the dynamics across time of lead and follow relations among groups. The algorithm takes as input the $w_{.,.,i}$ matrices for two or more factions, such as *Brexiteers* ($i = Brex.$) and *Remainers* ($i = Rem.$), each of which can be seen as a 38-dimensional time series (at the daily frequency), and, through a DTM algorithm, it gives as the output a dynamic directed and weighted network, for which nodes represent the factions and edges represent following relations between them. Each frame of this dynamic network represents a day. For each frame, inferred lead-follow relations between pairs of nodes are mutually exclusive; therefore, either *Brexiteers* follow *Remainers* ($Brex. \rightarrow Rem.$ and $Brex. \not\leftarrow Rem.$) or *Remainers* follow *Brexiteers* ($Brex. \not\rightarrow Rem.$ and $Brex.\leftarrow Rem.$) or no lead-follow relation is inferred ($Brex. \not\rightarrow Rem.$ and $Brex. \not\leftarrow Rem.$). $f_{i,j,\,x} \in [0, 1]$ is the weight of the edge $ij$ at the date $x$, and represents the strength of the follow relation (if any) between node $i$ and $j$ at a specific day. The values of $f_{Brex.,\,Rem.,\,x}$ and $f_{Rem.,\,Brex.,\,x}$ were used to build a partisan faction *lead/follow relation index* (central plot in Fig 8), which is

defined as follows:

$$fl_{Rem.,Brex.,x} = f_{Rem.,Brex.,x} - f_{Brex.,Rem.,x} \in [-1, 1] \tag{6}$$

To implement the FLICA algorithm, the mFLICA function from the mFLICA library for R [29] was employed, using a *window* of one month (30 days), a *max lag window* of six (6) days, and a *window time shift* of one (1) day. The results were found to be robust, and small and medium changes in the aforementioned parameters gave similar results.

## Supporting information

**S1 File.**
(PDF)

## Acknowledgments

The authors thank Manoel Horta Ribeiro and an anonymous reviewer for their useful suggestions and remarks, which helped improving the quality and clarity of this work.

## Author Contributions

**Conceptualization:** Carlo Romano Marcello Alessandro Santagiustina, Massimo Warglien.

**Data curation:** Carlo Romano Marcello Alessandro Santagiustina.

**Formal analysis:** Carlo Romano Marcello Alessandro Santagiustina, Massimo Warglien.

**Investigation:** Carlo Romano Marcello Alessandro Santagiustina, Massimo Warglien.

**Methodology:** Carlo Romano Marcello Alessandro Santagiustina, Massimo Warglien.

**Project administration:** Carlo Romano Marcello Alessandro Santagiustina, Massimo Warglien.

**Software:** Carlo Romano Marcello Alessandro Santagiustina.

**Supervision:** Carlo Romano Marcello Alessandro Santagiustina, Massimo Warglien.

**Validation:** Carlo Romano Marcello Alessandro Santagiustina, Massimo Warglien.

**Visualization:** Carlo Romano Marcello Alessandro Santagiustina.

**Writing – original draft:** Carlo Romano Marcello Alessandro Santagiustina, Massimo Warglien.

**Writing – review & editing:** Carlo Romano Marcello Alessandro Santagiustina, Massimo Warglien.

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
