## [Decision Letter · Decision Letter 0]

11 Jan 2022

PONE-D-21-32987The architecture of partisan debates: The online controversy on the no-deal BrexitPLOS ONE

Dear Dr. Santagiustina,

Thank you for submitting your manuscript to PLOS ONE. After careful consideration, we feel that it has merit but does not fully meet PLOS ONE’s publication criteria as it currently stands. Therefore, we invite you to submit a revised version of the manuscript that addresses the points raised during the review process.

In agreement with the reviewers' comments I recommend to focus a particular attention on contextualization of the paper in the state of the art, on figures' quality and on a more careful explanation of the methodologies. For this last point please keep in mind the interdisciplinary scope of the journal and make sure that the methodologies are well explained for a public potentially larger than for a disciplinary journal. 

We look forward to receiving your revised manuscript.

Kind regards,

Floriana Gargiulo

Academic Editor

PLOS ONE

“The authors acknowledge financial support from the European Union Horizon 2020 670

projects ODYCCEUS (Grant Agreement No. 732942) and ISEED (Grant Agreement 671

No. 960366).”

“C.S. and M.W. acknowledge financial support from the European Union Horizon 2020 projects ODYCCEUS (Grant Agreement No. 732942, website: www.odycceus.eu) and ISEED (Grant Agreement No. 960366, website: iseedeurope.eu). The funders  didn't play any role in the study design, data collection and analysis, decision to publish, or preparation of the manuscript.”

Reviewers' comments:

Reviewer's Responses to Questions

**Comments to the Author**

1. Is the manuscript technically sound, and do the data support the conclusions?

Reviewer #1: Yes

Reviewer #2: Yes

2. Has the statistical analysis been performed appropriately and rigorously? 

Reviewer #1: Yes

Reviewer #2: Yes

3. Have the authors made all data underlying the findings in their manuscript fully available?

Reviewer #1: Yes

Reviewer #2: No

4. Is the manuscript presented in an intelligible fashion and written in standard English?

Reviewer #1: Yes

Reviewer #2: Yes

5. Review Comments to the Author

Reviewer #1: This is a thoughtful and well-executed paper characterizing the debate on a no-deal Brexit. I congratulate the authors: the paper is well written, the analyses are methodologically sound, and the discussion is insightful and backed by the analyses in the paper. That being said, I believe there are some minor issues with the paper that would make it more fitting for publication. I think those changes can be executed easily, so I'm recommending a minor revision.

- I have found the quality of images to be below the acceptable standards for scientific publication. Figures 1 and 2 should have legends explaining the meaning on the line in the Figure (and not only in the caption). Figures and tables in general should be .pdfs or high quality images (for tables, you can generate the .pdf with a variety of tools, I recommend https://app.diagrams.net/).

- I appreciated the usage of STMs and the methodology to create the topic network in Figure 5. But the effect sizes on the differences captured by the edge colors are unclear to me. Authors should indicate the effect size either in the figure (at least for the discussed edges) or in the text (as they discuss bigrams).

- In Figure 6 I found the way in which you explained the topic ordering on the y-axis confusing. In my view it would make things clearer if you clarified that these correlations can be obtained in a STM by construction.

- I didn't get what additional insights came from Figure 7. This in my opinion is a weak result that did not add much to the paper. I get that you show there is assortativity between topics (which is already shown in 6) and between groups (i.e, Brexiteers vs. Remainers, which is new) —but wouldn't it be much easier to simply re-do Figure 6 for the groups?

Reviewer #2: In this work the authors analyse the partisan online debate on the no-deal Brexit. They focus in particular on Twitter. They analyse the structure of the debate and the main argumentations brought by the two factions of Breexiters and Remainers through Structural Topic Modeling.

Among the findings they identify specific caracteristics of the debate sustained by brexiters and Remainers: on the one hand Brexiters stressed the importance of topics like the greenfield trading opportunities and the increased authonomy. On the other hand Remainers focused more on the negative effects of a no-deal, among which hard border issues in Ireland and Scotland and helthcare problems.

I think the work is well presented and well organized. The authors provided a detailed and easy to read methodology, sufficient to allow their analysis to be reproduced.

From my point of views there are though some aspects that could be improved to make the work more impactfull. My suggestions mainly concern two aspects: 1) a better insertion of work in the scientific context and 2) the discussion of the limitations of work.

1) Concerning the need to better insert the work in the reference literature, I invite the authors to enrich the bibliography and to justify many of the statements present along the work, especially in the introduction, which at the moment do not find support in the bibliography.

Some examples:

> line 94-96: even though it’s clear the “no-deal” debate played a crucial role in the 2019 UK elections, citing some litterature would make the statement stronger

> line 22-24: “Public debates […] are strategic discussions about socially relevant issues.” Did somebody use this definition before? Is it always the case? Are we always strategic when we write online? I woun’t say so

> line 35-36: “These can be related...them”. There are so many works on the endogenous and exogenous effects on online attention dynamics. I report here few of them:

• Crane, R., & Sornette, D. (2008). Robust dynamic classes revealed by measuring the response function of a social system. Proceedings of the National Academy of Sciences, 105(41), 15649-15653

• Leskovec, J., Backstrom, L., & Kleinberg, J. (2009, June). Meme-tracking and the dynamics of the news cycle. In Proceedings of the 15th ACM SIGKDD international conference on Knowledge discovery and data mining (pp. 497-506).

> line 93: “ the no-deal aspect has polarised public opinions and received great media attention”. Many works have been written about polarization during the Brexit debate. Cite some!

• Del Vicario, M., Zollo, F., Caldarelli, G., Scala, A., & Quattrociocchi, W. (2017). Mapping social dynamics on Facebook: The Brexit debate. Social Networks, 50, 6-16.

• Hobolt, S. B., Leeper, T. J., & Tilley, J. (2021). Divided by the vote: Affective polarization in the wake of the Brexit referendum. British Journal of Political Science, 51(4), 1476-1493.

> line 47: “The proposed framework builds on existing litterature”. I think that it would be better to clearly state the previous works here, in order to give the expert reader some clear references. So instead of citing [22], [23], [24] in the Method section for the first time, I would suggest to anticipate them in the itroduction to give a clearer context to the reader.

2) Concerning the discussion of the limitations I would suggest to avoid some unjustified generalizations and I would like the authors to better discuss the limitations of their work.

Two points should be discussed in my opinion:

> The limit of studying the public debate on Twitter. Studying Twitter does not imply studying society. I would like the authors to stress in the discussion that anayzing the pubblic debate on twitter could be profoundly different from studying the public debate on the no-deal Brexit in general.

• Gayo-Avello, D. (2012). No, you cannot predict elections with Twitter. IEEE Internet Computing, 16(6), 91-94.

• Caldarelli, G., Chessa, A., Pammolli, F., Pompa, G., Puliga, M., Riccaboni, M., & Riotta, G. (2014). A multi-level geographical study of Italian political elections from Twitter data. PloS one, 9(5), e95809.

> I would avoid the first sentence of the Discussion: “ The present work shows that the structure and dynamics of online debates connect arguments in a coherent way across different levels and roles”. Is the online debate always rational and coherent? The authors identified 361116 relations on over 9 million Tweets. It seems that the majority of the tweets does not contain causal argumentations… I would like to read more about possible interpretations of why only few tweets have structured argumentations.

6. PLOS authors have the option to publish the peer review history of their article (what does this mean?). If published, this will include your full peer review and any attached files.

Reviewer #1: **Yes: **Manoel Horta Ribeiro

Reviewer #2: No

---

## [Author Response · Author response to Decision Letter 0]

25 May 2022

All responses are contained in the Response to Reviewers.pdf file.

---

## [Editor Report · Decision Letter 1]

7 Jun 2022

The architecture of partisan debates: The online controversy on the no-deal Brexit

PONE-D-21-32987R1

Dear Dr. Santagiustina,

We’re pleased to inform you that your manuscript has been judged scientifically suitable for publication and will be formally accepted for publication once it meets all outstanding technical requirements.

Kind regards,

Floriana Gargiulo

Academic Editor

PLOS ONE